# Competence of healthcare professionals in diagnosing and managing obstetric complications and conducting neonatal care: a clinical vignette-based assessment in district and subdistrict hospitals in northern Bangladesh

Abdullah Nurus Salam Khan, [1,2] Farhana Karim, [2]
Mohiuddin Ahsanul Kabir Chowdhury, [2,3] Nabila Zaka, [4] Alexander Manu, [5]
Shams El Arifeen, [2] Sk Masum Billah [2]

For numbered affiliations see end of article.

**Correspondence to**
Mr Sk Masum Billah;
billah@icddrb.org

## ABSTRACT

**Background** This study assesses the competency of maternal and neonatal health (MNH) professionals at district-level and subdistrict-level health facilities in northern Bangladesh in managing maternal and newborn complications using clinical vignettes. The study also examines whether the professional's characteristics and provision of MNH services in health facilities influence their competencies.

**Methods** 134 MNH professionals in 15 government hospitals were interviewed during August and September 2016 using structured questionnaire with clinical vignettes on obstetric complications (antepartum haemorrhage and pre-eclampsia) and neonatal care (low birthweight and immediate newborn care). Summative scores were calculated for each vignette and median scores were compared across different individual-level and health facility-level attributes to examine their association with competency score. Kruskal-Wallis test was performed to identify the significance of association considering a p value<0.05 as statistically significant.

**Results** The competency of MNH professionals was low. About 10% and 24% of the health professionals received 'high' scores (>75% of total) in maternal and neonatal vignettes, respectively. Medical doctors had higher competency than nurses and midwives (score=11 vs 8 out of 19, respectively; p=0.0002) for maternal vignettes, but similar competency for neonatal vignettes (score=30.3 vs 30.9 out of 50, respectively). Professionals working in health facilities with higher use of normal deliveries had better competency than their counterparts. Professionals had higher competency in newborn vignettes (significant) and maternal vignettes (statistically not significant) if they worked in health facilities that provided more specialised newborn care services and emergency obstetric care, respectively, in the last 6 months.

**Conclusions** Despite the overall low competency of MNH professionals, exposure to a higher number of obstetric cases at the workplace was associated with their competency. Arrangement of periodic skill-based and

### Strengths and limitations of this study

► The study assesses the competence (knowledge and skill) of doctors, nurses and midwives in managing complicated obstetric and neonatal cases using clinical vignettes.
► The study identifies the association of healthcare professionals' competency with individual-level factors, such as cadre type, work experience and training status, and health facility-level factors, such as health facility type, provision of maternal and neonatal health services, and obstetric caseload.
► The study highlights the need for local and national health authorities to prioritise continuous professional development and practical clinical training to ensure a fully competent and skilled birth attendance in low-resource settings.
► The study does not validate the competency assessed by clinical vignettes by comparing it with direct clinical observation or chart abstraction of performed services at clinical settings.

drill-based in-service training for MNH professionals in high-use neighbouring health facilities could be a feasible intervention to improve their knowledge and skill in obstetric and neonatal care.

## BACKGROUND

Maternal and neonatal survival has received significant priorities in the global agenda of Millennium Development Goals (MDGs) and subsequently in Sustainable Development Goals.[1–3] During the MDG era, the maternal mortality ratio (MMR) and the neonatal mortality rate (NMR) have been reduced by 44% and 49%, respectively, between 1990 and 2015.[4 5] Despite the progress achieved in

reducing MMR and NMR, 303 000 mothers and 4 million newborns are still estimated to be dying each year worldwide, and 99% of them are dying in low-income regions of the world.[4 5] A WHO systematic review finds that around 27% and 14% of all maternal deaths are due to haemorrhage and complications arising from hypertensive disorders of pregnancy, including pre-eclampsia, respectively.[6] For newborns, the major reasons for mortality are preterm birth, birth asphyxia and infections.[7 8] The evidence indicates that the highest mortality risk for both mothers and their babies is on the first day of birth.[9] It is estimated that 16%–33% of maternal mortality can be averted if the major causes of death can be managed timely and efficiently by skilled attendance during childbirth.[10] The mere presence of a skilled birth attendant in health facilities equipped with key signal functions of emergency obstetric care, however, does not ensure the best practices of intrapartum and immediate postpartum care. To improve the survival of mothers and newborns through the best practices, health professionals' competence to diagnose and manage complications arising at the time of childbirth is strongly required as low-income and middle-income countries (LMICs) are struggling to increase the skilled birth attendance for every childbirth.[11 12] The knowledge and skill gap of the skilled birth attendants in LMICs often remain unaddressed as the countries have limited resources for assessing and improving the competencies of maternal and neonatal health (MNH) professionals.[13 14]

A clinical vignette or a written case simulation is an inexpensive and validated way to reflect on the performance and quality of healthcare professionals' actual practice,[15 16] especially in low-income settings where chart documentation is often incomplete for obstetric cases.[17 18] Assessment based on clinical vignettes is more feasible and accurate than direct clinical observation of health professionals' performance where actual performance could be limited by the shortage of essential drugs, equipment or supplies, and infrequent presentation of critical obstetric cases (<15% of all pregnancies).[19 20] This paper aimed to assess the competency of healthcare professionals at district-level and subdistrict-level health facilities in northern Bangladesh in managing maternal and newborn complications using clinical vignettes. The clinical vignettes are adopted from two studies conducted in Ghana as part of the overall quality of care evaluation of hospital-based MNH services.[21 22] The authors found overall moderate competency of MNH workers and its association with demand for facility deliveries, availability of infrastructure and workload, but did not explore the association with different attributes of healthcare professionals in detail. Independent of focusing on another LMIC context, this paper also explores whether individual-level factors, such as cadre type, training on emergency obstetric and neonatal care, and work experience, and health facility-level factors, such as health facility type, obstetric caseloads (ie, annual delivery conducted in health facilities), and provision of emergency obstetric and neonatal care, are associated with the competence of the health professionals. Findings from this paper will be useful in designing capacity development programmes for MNH professionals and in bringing procedural changes in health facilities of LMICs to improve MNH outcomes through evidence-based skilled childbirth services.

## METHODOLOGY
### Study design and study sites
This study draws on the findings from a cross-sectional assessment conducted for the baseline evaluation of 'Every Mother Every Newborn Quality Improvement' initiatives, jointly implemented by Government of Bangladesh, UNICEF headquarter and UNICEF Bangladesh. The initiative focuses on implementing quality of care standards for maternal and neonatal healthcare delivery in the selected health facilities of three neighbouring districts—Kurigram, Lalmonirhat and Gaibandha—situated in the northern part of Bangladesh. From each of the districts, four subdistrict health facilities or upazila health complex (UHC) and one district hospital (DH) were selected. All UHCs are primary-level referral facilities designated to provide at least basic emergency obstetric and newborn care (EmONC). The DHs, secondary level health facilities, offer comprehensive EmONC services. The data were collected between August and September 2016.

### Selection of MNH care professionals
From 3 DHs and 12 UHCs, a total of 134 healthcare professionals who were posted and present at these health facilities on the assessment day, and were responsible to conduct normal deliveries and manage obstetric or neonatal emergencies were selected for the assessment. The 134 respondents included 39 medical doctors, 91 nurses and four midwives.

### Administration of vignettes
We used two vignettes (case A and case B) to describe obstetric complications and two vignettes (case C and case D) on newborn care. Case A and B describe scenarios on pregnant women with signs of pre-eclampsia and antepartum haemorrhage (APH), respectively, and both cases include separate sections on 'diagnosis' and 'management' of those conditions. Case C describes a baby with foetal distress and comprises three parts: (1) resuscitation, (2) immediate newborn care and (3) thermal care. Case D is about very low birthweight babies and comprises two parts: (1) immediate management and (2) counselling caregivers on breast feeding.

A team of trained research physicians administered all four clinical vignettes among the health professionals through face-to-face interview using a structured questionnaire including a checklist. At first, the research physician read out the scenarios to the healthcare professionals and asked for specific actions to be taken for each of the scenarios. It was done one scenario at a time up to where

the action was required. Responses made by the health professionals to each of the scenarios were then matched with the predefined vignette action points in the checklist (online supplementary appendix S1). The vignette action points were based on the WHO's Integrated Management of Pregnancy and Childbirth guidelines for the selected obstetric and neonatal cases.[23] Responses were considered as correct if they were in agreement with that in the list. The research physicians were trained on the vignette scenarios and the definition of each vignette action points. Since all four vignette scenarios were read out and responses were collected for each, there was no incomplete data. Intraobserver and interobserver variabilities in recording the responses were less likely to occur as the responses were structured.

### Assessing the provision of MNH services

In 15 health facilities, a separate team of research assistants extracted the information on the number of conducted deliveries and the provision of emergency obstetric care and specialised newborn services in the last 6 months from health facility records using a checklist. Emergency obstetric care included administration of injectable antibiotics, injectable oxytocin and injectable anticonvulsant; manual removal of retained placenta and retained products of conception; and provision of assisted vaginal delivery, blood transfusion, and caesarean delivery. Specialised newborn services were the provision of kangaroo mother care (KMC), basic resuscitation (clearing of the airway, stimulation and bag–mask ventilation), advanced resuscitation (intubation and chest compression), administration of oxygen, intravenous fluid and antenatal corticosteroid.

### Data analysis

To analyse the total vignette score obtained by the respondents, each of the 'actions' was assigned to either equal score (one point for each action point listed for cases A and B) or weighted scores (for cases C and D). The detailed methodology of assigning the scores are mentioned elsewhere,[21 22] and the exact scores for each action points used in analysis are listed in online supplementary appendix S1. The authors mentioned in their papers that they had asked expert opinions on the relative importance of the actions for all four vignette cases to assign weighted scores for each action point. Based on the post hoc analysis, the authors decided to keep the simpler approach of assigning equal score for maternal vignettes (cases A and B) as there were no major differences in results between equal or weighted scores,[21] but chose weighted scoring for neonatal vignettes (cases C and D).[22]

Based on the assigned scores, summative scores were calculated for the vignette cases on maternal and neonatal complications. Vignette scores were compared between different categories of individual-level and health facility-level attributes using Kruskal-Wallis tests. The scores were further categorised into 'high' if the health professional received more than 75% of the total score, 'moderate' for 50%–75% of the total and 'low' for less than 50% of the total. Healthcare professionals were categorised based on their cadre types, total working experience, training status, and whether they were placed in a maternal or neonatal unit in the health facility; and health facilities were categorised based on their type (district or subdistrict), annual caseload of deliveries and provision of emergency management for maternal and neonatal complications. The average number of annual deliveries conducted in the health facilities is presented as mean with SD. Based on the average caseload of deliveries, the health facilities were further categorised into 'low' use (if they conducted less than 600 deliveries a year on an average), 'medium' use (600–900 deliveries) and 'high' use (more than 900 deliveries a year). The cut-off points were set roughly matching with the expected number of institutional births in EmONC equipped health facilities based on population size and expected number of pregnancies in Bangladesh[24]; however, the range does not represent the national or regional estimate of institutional deliveries in DHs or sub-DHs. Health facility-level service provision indicators were considered 'available' only if the activities related to these indicators were performed within 6 months prior to the assessment and reported using frequency and percentage of the health facilities providing them. A p value less than 0.05 was considered as statistically significant for the Kruskal-Wallis test. The analysis was performed using Stata V.13.0.

### Patient and public involvement

The research questions and outcome measures were related to assessing the knowledge and skill of government healthcare professionals in managing obstetric and neonatal complications using clinical vignettes. In the selection of cases for clinical vignettes, the critical cases were chosen and presented as they would occur among pregnant women and their newborn in the context of Bangladesh. Patients or pregnant women were not involved in the design of, the recruitment to and the conduct of the study. The baseline results were disseminated among the health managers and MNH professionals in the government hospitals where the study was conducted, and the endline results will be disseminated in due course.

## RESULTS
### Facility delivery use and provision of emergency obstetric and specialised neonatal care

Table 1 presents the status of delivery use and service provision of the surveyed health facilities regarding emergency obstetric and newborn care by health facility types, district and subdistrict. During the preceding 3 years since the assessment, the average number of deliveries conducted in a year was less than 600 (low use) in 10 out of 12 sub-DHs, whereas in two out of three DHs and in one sub-DH, the average number was more than 900 a year

**Table 1** Annual delivery use and provision of emergency obstetric and specialised neonatal care in the health facilities within 6 months prior to the assessment

| Indicators | Total, n (%) | District hospital, n (%) | Subdistrict hospital, n (%) |
|---|---|---|---|
| Total number | 15 | 3 | 12 |
| Annual delivery conducted on average | | | |
| Mean (±SD) | 509 (±335) | 908 (±301) | 410 (±268) |
| Low, <600 | 10 (66.7) | 0 (0.0) | 10 (83.3) |
| Medium, 600–900 | 2 (13.3) | 1 (33.3) | 1 (8.3) |
| High, >900 | 3 (20.0) | 2 (66.7) | 1 (8.3) |
| Provision of emergency obstetric care | | | |
| Injectable antibiotics administration | 15 (100.0) | 3 (100.0) | 12 (100.0) |
| Injectable oxytocin administration | 15 (100.0) | 3 (100.0) | 12 (100.0) |
| Injectable anticonvulsant administration | 8 (53.3) | 3 (100.0) | 5 (41.7) |
| Manual removal of retained placenta | 14 (93.3) | 3 (100.0) | 11 (91.7) |
| Manual removal of retained products of conception | 14 (93.3) | 3 (100.0) | 11 (91.7) |
| Assisted vaginal delivery | 0 (0.0) | 0 (0.0) | 0 (0.0) |
| Blood transfusion services | 6 (40.0) | 3 (100.0) | 3 (25.0) |
| Caesarean delivery | 4 (26.7) | 3 (100.0) | 1 (8.3) |
| Provision of specialised care for newborn | | | |
| Provision of kangaroo mother care | 0 (0.0) | 0 (0.0) | 0 (0.0) |
| Basic resuscitation* | 15 (100.0) | 3 (100.0) | 12 (100.0) |
| Advanced resuscitation† | 4 (26.7) | 2 (66.7) | 2 (16.7) |
| Oxygen administration | 15 (100.0) | 3 (100.0) | 12 (100.0) |
| Intravenous fluid administration | 14 (93.3) | 3 (100.0) | 11 (91.7) |
| Antenatal corticosteroid administration | 3 (20.0) | 2 (66.7) | 1 (8.3) |

*Clearing of airway, stimulation, and bag–mask ventilation.
†Intubation and chest compression.

(high use) and 2 of the 15 health facilities were between 600 and 900 a year (medium use). On average, the DHs had higher obstetric caseloads than the sub-DHs (908±301 vs 410±268 annual deliveries). All signal functions of comprehensive emergency obstetric care but 'assisted vaginal delivery' were performed in all three DHs within 6 months prior to the assessment. Injectable anticonvulsant and blood transfusion were provided and caesarean deliveries were performed in 42%, 25% and 8% of the 12 sub-DHs, respectively. KMC was not practised in any of the health facilities. For birth asphyxia management, basic resuscitation procedures, such as airway clearing, stimulation and bag–mask ventilation, were practised in all the health facilities within 6 months prior to the assessment, but advanced resuscitation measures, such as intubation or chest compression, were only conducted in two district and two sub-DHs within the same period.

### Characteristics of healthcare professionals

The mean (±SD) age of 134 health professionals was 37.7 (±7.5) years; however, the mean age of nurses and midwives was higher than that of medical doctors (40.0±6.9 vs 32.2±6.1 years). About one-third of doctors

and all the nurses and midwives were female (table 2). Among the interviewed health professionals, 84% and 62% were working in maternity and neonatal wards, respectively. For half of the health professionals, both working areas overlapped as they had experience of working in both wards. None of the medical doctors had more than 5 years of working experience, whereas 59% of the nurses and midwives had so. A higher proportion of nurses and midwives had ever received an in-service training on obstetric complications compared with the doctors (96% vs 44%), and about 45% of all health professionals had ever received an in-service training on newborn complications. Within 12 months prior to the assessment, only 7% and 29% of health professionals received any training on obstetric and newborn complications, respectively.

### Competence of health professionals in diagnosing and managing maternal and neonatal complications

For vignettes dealing with obstetric complications (cases A and B), the median summative score was 8 with an IQR of 6–11 (table 3). About 60% of the health professionals received a low score, that is, less than 50% of the maximum

**Table 2** Characteristics of health professionals

| Characteristics | Total (%) | Medical doctor (%) | Nurse and midwife (%) |
|---|---|---|---|
| Total number | 134 | 39 | 95 |
| Age (years) | | | |
| Less than 30 | 21.6 | 59.0 | 6.3 |
| 31–40 | 48.5 | 33.3 | 54.7 |
| 41–50 | 21.6 | 2.6 | 29.5 |
| More than 50 | 8.2 | 5.1 | 9.5 |
| Female health professionals | 79.9 | 30.8 | 100.0 |
| Type of health facility where posted | | | |
| District hospital | 24.6 | 17.9 | 27.4 |
| Subdistrict hospital | 75.4 | 82.1 | 72.6 |
| Majorly works in | | | |
| Both maternity and neonatal wards | 51.5 | 56.4 | 49.5 |
| Maternity ward | 83.6 | 87.2 | 82.1 |
| Neonatal ward | 61.9 | 61.5 | 62.1 |
| Work experience (years) | | | |
| Less than 1 | 8.2 | 17.9 | 4.2 |
| 1–2 | 38.8 | 71.8 | 25.3 |
| 3–5 | 11.2 | 10.3 | 11.6 |
| More than 5 | 41.8 | 0.0 | 58.9 |
| Ever received in-service training on | | | |
| Management of obstetric complication | 80.6 | 43.6 | 95.8 |
| Management of newborn complication | 44.8 | 51.3 | 42.1 |
| Received training in the last 12 months on | | | |
| Management of obstetric complication | 6.7 | 5.1 | 7.4 |
| Management of newborn complication | 29.1 | 28.2 | 29.5 |

possible score, and only 10% achieved a high score (more than 75% of the total score). Between the two obstetric cases, health professionals had slightly better knowledge about diagnosis and management of eclampsia (case A) than APH (case B) (49% received moderate to high scores vs 38%, respectively). Moreover, 51% of the health professionals had moderate to high scores regarding the diagnosis of APH and eclampsia in comparison with 36%, who had a similar competency regarding the management of these cases.

The median summative vignette score for neonatal complications (cases C and D) was 30.8 (IQR 23.9–36.6), where 50 was the maximum possible score. About 70% of the health professionals could achieve at least 50% of the total score in newborn vignettes. Between the two cases, health professionals' competence on provision of essential newborn care (case D) was better than that for management of very low birthweight babies (case C), as evident from the categorical classifications of scores (78% received moderate to high scores vs 58%, respectively).

Table 4 presents the distribution of median vignette scores for maternal and neonatal complications across different attributes of the health professionals and their working health facilities. For maternal complications, the median scores differed significantly between the two cadres of health professionals (p=0.0002), with medical doctors achieving a higher score than nurses or midwives. Neither of the maternal or neonatal vignette scores were significantly different among the health professionals who worked in both maternal and neonatal wards or only in one of these wards or between two levels of health facilities. Health professionals' duration of work experience or their receipt of training on management of obstetric or neonatal complication did not make any significant difference in their competencies. Health professionals working in high-use health facilities with more than 900 annual deliveries had significantly better competency in diagnosing and managing both maternal and neonatal complications than their counterparts (p=0.0005 and p=0.004, respectively). Health professionals working in health facilities that provided more than five out of seven EmOC services had statistically non-significant (p=0.6819) but higher competency in maternal vignettes. Similarly, health professionals had a significantly higher competency in newborn vignettes (p=0.0361) if they were working in health facilities that were better equipped to provide emergency management for newborn complications.

### Competency of healthcare professionals by obstetric caseloads of health facilities

Figures 1 and 2 show the vignette scores for obstetric and neonatal complications, respectively, across the two cadres of health professionals by the health facility's annual delivery use. The competence of the health professionals working in low-use health facilities fall majorly in 'low' to 'moderate' categories, and the median scores do not differ substantially between medical doctors and nurses or midwives. Medical doctors working in high-use health facilities show 'high' competency compared with most nurses or midwives, especially in newborn complication management.

### DISCUSSION

The study finds low competency of diagnosing and managing critical obstetric and neonatal conditions among the health professionals at DHs and sub-DHs in northern Bangladesh. Inadequate provision of health services for management of obstetric emergencies or specialised newborn care in the health facilities, along with low coverage of recent refresher training, might have reflected in low competencies of the MNH professionals at these hospitals. The competency, as measured by the

**Table 3** Median vignette scores and their categorical classifications

| Vignette cases | Total range | Median score (IQR) | Categorical score (%), N=134 | | |
| --- | --- | --- | --- | --- | --- |
| | | | High (>75%) | Moderate (50%–75%) | Low (<50%) |
| **Maternal complication** | | | | | |
| Case A (eclampsia) | 0–09 | 4 (3–6) | 12.7 | 36.6 | 50.7 |
| Case B (antepartum haemorrhage) | 0–10 | 4 (2–6) | 8.2 | 29.9 | 61.9 |
| Case A+B (diagnosis component) | 0–09 | 5 (3–6) | 11.2 | 40.3 | 48.5 |
| Case A+B (management component) | 0–10 | 4 (2–6) | 8.2 | 27.6 | 64.2 |
| Case A+B (total) | 0–19 | 8 (6–11) | 10.4 | 29.9 | 59.7 |
| **Neonatal complication** | | | | | |
| Case C (care for VLBW babies) | 0–20 | 11.1 (8.4–14.9) | 23.9 | 34.3 | 41.8 |
| Case D (essential newborn care) | 0–30 | 19.8 (15.6–24.4) | 30.6 | 47.0 | 22.4 |
| Case C+D (total) | 0–50 | 30.8 (23.9–36.6) | 23.9 | 46.3 | 29.9 |

IQR, Interquartile range; VLBW, very low birth weight.

vignette scores, varied significantly between the cadres of health professionals and was associated with obstetric caseloads of the health facilities.

All healthcare professionals were working in obstetric or neonatal wards in the selected health facilities and were expected to achieve standard competencies for the first-line management of the scenarios presented in both the maternal and the newborn vignettes.[24] The vignette scores indicate that majority of the professionals achieved less than 50% of the total score, that is, low competency for the obstetric complications (APH and eclampsia), but showed better competencies for newborn complications. This could result from the poor retention of knowledge of managing obstetric complication due to lack of recent training on this topic. The data show that the proportion of health professionals receiving a training within the last 12 months on the management of maternal complication was lower than those receiving training on neonatal complications (7% vs 30%), whereas most of them had ever received in-service training on these topics. Although capacity development through targeted training of the maternal, newborn and child health (MNCH) care professionals is identified as a critical intervention to improve health outcome,[25] this often fails to meet the desired outcome due to inadequate post-training monitoring, follow-up and periodic refresher training.[26] One study in India tested the feasibility and effectiveness of skill-and-drill based training for MNCH workers at subdistrict-level and district-level health facilities and found improved knowledge and skill of the health workers following the intervention.[27] The authors, however, emphasised on bringing system-level improvement, that is, ensuring appropriate infrastructure and equipment to translate the improved competency into actual practices. The study also identified that both vignette scores were higher among the health professionals who worked in health facilities with a 'high' caseload of conducting deliveries but remained nearly similar in low-use and medium-use

health facilities. This association further underscores the fact that experiential learning or on-the-job experiences can contribute to the competence achieved through academic training only.[28 29] The comprehensive package of training for EmONC could, therefore, include skill-based and drill-based in-service training. The same health facility where the health professionals are posted or a neighbouring high-use health facility can serve as an 'obstetric-skill laboratory' or a 'simulation centre' for conveying this type of training periodically.[30] Operations researches should further explore effective and efficient ways to incorporate such interventions into the government's existing EmONC training programmes in low-resource settings of the LMICs.

Our study also identifies that the doctors had significantly better knowledge scores than nurses and midwives in maternal complication vignettes but similar scores in newborn vignettes. With the higher extent of academic and clinical training, it is not surprising that doctors would have higher competency scores than nurses and midwives in the maternal vignettes, which, in general, require more intensive clinical procedures than the newborn scenarios. Their similar competence in newborn vignettes, however, indicates that advanced preservice clinical training of the health professionals does not necessarily translate into better knowledge and skill, and similar evidence was found for sick child management in a multicountry study.[31] This directs the policymakers to consider the role of task shifting from doctors to nurses and midwives to maximise the productive efficiency in health facilities of LMICs, particularly for low intensive procedures like immediate newborn care or identification of obstetric complications and referral.[32 33] This calls for a timely initiative as in Bangladesh the newly deployed midwifery cadre is expected to work with its full capacity from 2016 and onwards, according to the Midwifery Act.[34]

The study found that median maternal vignette scores were higher, although statistically not significant, among

**Table 4** Median scores of maternal and neonatal vignettes across the categories of individual-level and health facility-level attributes

| Characteristics | Maternal complication (cases A and B) | | Neonatal complication (cases C and D) | |
|---|---|---|---|---|
| | Median score* (IQR) | P value† | Median score* (IQR) | P value† |
| **Type of health professionals** | | | | |
| Medical doctor | 11 (7–14) | **0.0002** | 30.3 (21.0–45.3) | 0.6682 |
| Nurse and midwives | 8 (6–10) | | 30.9 (25.5–35.5) | |
| **Majorly works in** | | | | |
| Both maternity and neonatal ward | 9 (6–11) | 0.7540 | 30.9 (24.1–36.0) | 0.3636 |
| Maternity ward only | 8 (6–11) | | 27.9 (23.0–36.6) | |
| Neonatal ward only | 8 (7–13) | | 32.3 (26.0–39.2) | |
| **Work experience (year)** | | | | |
| Less than 1 | 10 (7–12) | 0.1099 | 30.9 (23.3–37.5) | 0.8443 |
| 1–5 | 9 (7–12) | | 31.6 (22.1–39.2) | |
| More than 5 | 7.5 (5–10) | | 29.7 (25.7–35.3) | |
| **Ever trained on management of obstetric complication** | | | | |
| Yes | 8 (6–11) | 0.0931 | 30.9 (24.5–36.1) | 0.6628 |
| No | 9 (7–13) | | 29.5 (20.5–45.3) | |
| **Ever trained on management of neonatal complication** | | | | |
| Yes | 8.5 (5–11) | 0.8005 | 30.8 (21.8–36.3) | 0.3817 |
| No | 8 (6–12) | | 30.8 (25.9–37.6) | |
| **Received training in the last 12 months on the management of obstetric complication** | | | | |
| Yes | 10 (6–12) | 0.5456 | 30.3 (27.6–36.3) | 0.3789 |
| No | 8 (6–11) | | 30.7 (23.3–36.6) | |
| **Received training in last 12 months on the management of neonatal complication** | | | | |
| Yes | 9 (6–12) | 0.3674 | 31.9 (23.0–37.5) | 0.5567 |
| No | 8 (6–11) | | 30.1 (24.3–36.6) | |
| **Type of health facility where posted** | | | | |
| District hospital | 8 (7–11) | 0.6682 | 28.7 (25.5–35.5) | 0.4618 |
| Subdistrict hospital | 8 (6–11) | | 31.3 (23.3–37.7) | |
| **Annual delivery conducted in the health facility** | | | | |
| Low, <600 | 8 (6–11) | **0.0005** | 30.0 (22.9–36.3) | **0.004** |
| Medium, 600–900 | 7 (5–9) | | 27.4 (23.3–34.3) | |
| High, >900 | 12 (9–15) | | 34.5 (30.2–46.1) | |
| **Provision of EmOC‡** | | | | |
| Five or less services | 8 (6–12) | 0.6819 | 31.3 (23.2–38.2) | 0.4164 |
| More than five | 9 (7–11) | | 29.1 (25.5–35.5) | |
| **Provision of emergency management for neonatal complication§** | | | | |
| Three or less services | 8 (6–11) | 0.2722 | 29.3 (23.0–35.7) | **0.0361** |
| More than three | 8 (7–12.5) | | 31.8 (26.0–38.7) | |

Boldfaced values are statistically significant.
*Score range for maternal complication is 0–19 and that for neonatal complication is 0–50.
†P value is obtained from the Kruskal-Wallis test.
‡Provision of injectable antibiotic, oxytocin and anticonvulsant; manual removal of placenta and retained products; and performance of blood transfusion and caesarean sections within 6 months prior to the assessment.
§Provision of kangaroo mother care; basic and advanced resuscitation; and administration of oxygen, intravenous fluids, and antenatal corticosteroids within 6 months prior to the assessment.
EmOC, emergency obstetric care; IQR, Interquartile range.

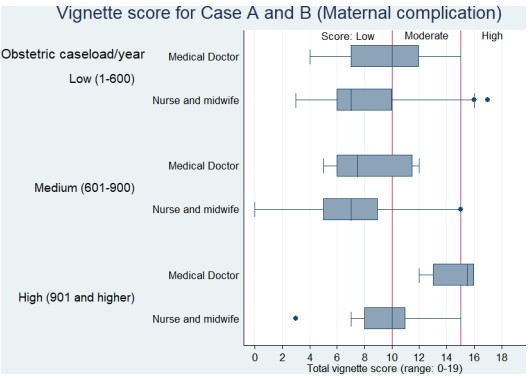

**Figure 1** Vignette scores for obstetric complications by health professional cadre and type of health facility.

the health professionals who worked in health facilities that were better equipped to provide emergency obstetric services, whereas for newborn vignettes, the scores were significantly higher in health facilities that could provide better newborn specialised services. This indicates that inadequate service provision for emergency management in the health facilities could result in low competency of the MNH professionals through infrequent exposure to critical cases. In the assessed sub-DHs, blood transfusion, which is required in case of severe APH or injectable magnesium sulfate for early stabilisation of pre-eclampsia, was infrequently provided in the last 6 months. The gap was also profound for conducting caesarean sections, providing advanced resuscitation for newborns and administering antenatal corticosteroid for premature labour. KMC, evidence-based care for low birthweight babies, was not provided in any of the assessed health facilities. The literature suggests that the competence of health professionals is shaped from actively engaging in evidence-based practices,[29] and to implement such practices, the readiness of the health facilities should be optimum for carrying out the intended procedures.[35 36] Improving the competence of healthcare professionals to ensure the procedural quality of care should, therefore, require structural improvement of the health facilities to allow them to conduct the required tasks.

The findings of the study should be cautiously interpreted as the study assesses the competence using clinical vignettes that explore what one should do at an expected level, not what one actually performs in real settings.[37] Clinical vignettes should be considered as a cheap alternative to direct clinical observation for assessing the knowledge and skill of health professionals or where chart abstraction is not feasible due to poor chart documentation of obstetric cases.[16] The study objectives do not also consider whether the competency gap of MNH professionals translate into poor quality of maternal and neonatal care in the selected health facilities. Further research should focus on validating the vignette-based findings by examining the association of health professionals' competency with their actual practices and the related health outcomes for mothers and newborns in the health facilities. The first-level health facilities are not included in the study as they are not equipped or entitled to provide EmONC services and monthly use of normal delivery is very low.[24] The study findings on health professionals' competence are therefore limited to district-level and subdistrict-level health facilities only.

## CONCLUSION

Clinical vignettes could be a useful tool to assess the competency, a key predictor of performance, of MNCH professionals in low-resource settings. Health professionals' cadre and obstetric caseload of the working health facilities were associated with their competence in diagnosing and managing maternal and newborn complications. The arrangement of periodic skill-based and drill-based in-service and refreshers training using high-use neighbouring health facilities as obstetric-skill laboratories could be a feasible intervention to improve the competence of MNCH professionals in LMICs.

**Author affiliations**
[1]Health Promotion, Education and Behavior, University of South Carolina, Columbia, South Carolina, USA
[2]Maternal and Child Health Division, International Centre for Diarrhoeal Disease Research Bangladesh, Dhaka, Bangladesh
[3]Epidemiology, University of South Carolina, Columbia, South Carolina, USA
[4]Health Section, Maternal and Newborn Health, UNICEF USA, New York, New York, USA
[5]Department of Population Health, Liverpool School of Tropical Medicine, Liverpool, Liverpool, UK

**Acknowledgements** We acknowledge the support received from the Quality Improvement Secretariat of the Ministry of Health and Family Welfare and UNICEF Bangladesh to facilitate the data collection. We thank the research participants, that is, healthcare professionals and hospital managers, for allowing us to conduct the study. We also thank the data collection and entry team for their commitment and sincere effort. icddr,b gratefully acknowledges the following donors who provide unrestricted support: Government of the People's Republic of Bangladesh; Global Affairs Canada (GAC); Swedish International Development Cooperation Agency (Sida) and the Department for International Development, (UKAid).

**Contributors** SEA, SMB, NZ and MAKC designed the study, and SMB, MAKC and ANSK conceptualised the paper. ANSK, FK and MAKC were involved in data acquisition and literature review. ANSK and FK conducted the analysis of the paper, and NZ, AM and SEA guided the additional analysis. ANSK and FK prepared the first draft. MAKC, NZ, AM, SEA and SMB contributed to the revision and preparation of the final draft. All authors have reviewed and approved the final manuscript.

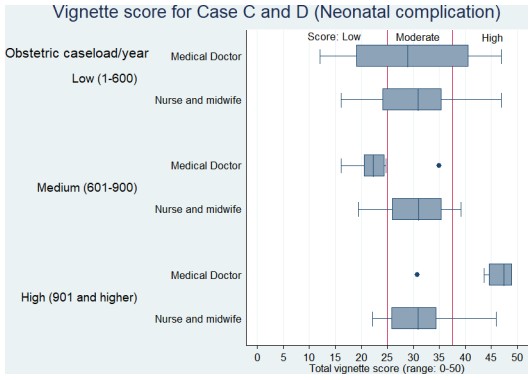

**Figure 2** Vignette scores for neonatal complications by health professional cadre and type of health facility.

**Funding** The study was conducted with funding support from Bill and Melinda Gates Foundation (BMGF) for Every Mother Every Newborn Quality Improvement initiative to UNICEF Headquarter (BMGF grant number OPP1112117, UNICEF and BMGF partnership grant).

**Competing interests** None declared.

**Patient consent for publication** Not required.

**Ethics approval** The ethical review committee of ICDDR,B approved the study (protocol number PR-16024). Informed written consent was sought from the healthcare professionals and health facility managers before conducting the interviews.

**Provenance and peer review** Not commissioned; externally peer reviewed.

**Data availability statement** Data are available upon reasonable request.

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
