## [Reviewer comments · BMJ Open]

ARTICLE DETAILS

TITLE (PROVISIONAL)	Competence of the Health Care Professionals in Diagnosing and Managing Obstetric Complications and Conducting Neonatal Care: A Clinical Vignette Based Assessment in District and Sub-District Hospitals in Northern Bangladesh
AUTHORS	Khan, Abdullah Nurus Salam; Karim, Farhana; Chowdhury, Mohiuddin Ahsanul Kabir; Zaka, Nabila; Manu, Alexander; Arifeen, Shams E.; Billah, Sk

VERSION 1 - REVIEW

REVIEWER	Dr. Nicholas N.A. Kyei Heidelberg Institute of Global Health University of Heidelberg Germany & 37 Military Hospital Accra, Ghana
REVIEW RETURNED	23-Jan-2019

GENERAL COMMENTS	Summary: This paper applies tailored clinical vignettes to assess the competencies of “skilled birth attendants” at primary and secondary health care facilities in northern Bangladesh to identify and manage some common life-threatening obstetric and newborn complications. Findings highlights the need for local and national health authorities to prioritize continuous professional development and practical clinical trainings to assure a fully competent and skilled birth attendance as we aim to end preventable maternal and newborn deaths in such low-resource settings. The authors list some strengths and limitations of this study, however authors would need to consider and address some of the following to improve the manuscript and make it publishable: Major changes Introduction: • Page 4 lines 76 – 79: in defining the role of a skilled birth attendant, authors omit the relevant reference (Making pregnancy safer: the critical role of the skilled attendant: a joint statement by WHO, ICM and FIGO. Geneva: WHO; 2004) that defines who a skilled birth attendant is and their required skills and abilities.• Page 3, Lines 80 -89: This section is verbose and the main argument is difficult to follow. Please make it more concise and clear
---

- Lines 87-88, authors state some pros for using clinical vignettes but do not mention other valid methods that have been used. For example, in; Harvey SA, Blandón YC, McCaw-Binns A, et al. Are skilled birth attendants really skilled? A measurement method, some disturbing results and a potential way forward. Bull World Health Organ. 2007;85(10):783-90. Reviewing this method may highlight some other limitations of a quick vignette system.

Methodology:

- Page 6, lines 125 -131; Were all four vignettes read out in full before the practitioner was made to answer? Or it was done one scenario at a time up to where action was required? Please provide a clearer description of how this was done
- Also make clear how the answers were matched as stated in lines 127-129
- Very little information is provided on how the overall scores were obtained. Although authors refer to two papers with a detailed description, sufficient information must be provided in this paper to enable readers to understand the main indicator of this manuscript without having to read two other papers.

Results:

- Page 8 line 174, since readiness and actual provision are not exactly the same, I recommend that you replace the readiness to provision. For example, a facility may be equipped to perform assisted vaginal deliveries and thus is technically ready but may never actually perform same.
- The number of facility deliveries is presented as an ordinal variable, however the basis of the categorization is unclear and not mentioned in the methods section. Furthermore, instead of stating that the average number of deliveries was less than 600 at a particular facility, it may be better to provide the actual average with standard deviation as well as the ranges for such a variable.
- Similarly, on Page 9, lines 195 -196: Better to provide the mean age with standard deviation and range
- Table 2, Page 9, line 206: It is unclear whether the numbers in the table represent absolute numbers or percentages. Please indicate in the column headings
- Page 10, lines 209 -216, It appears also that the vignettes scores were categorized as an ordinal variable. However, no mention of this of the basis is provided in the methods section.

Minor changes

Introduction:

- Page 4 line 67 “A WHO systematic review shows around 27% dies of haemorrhage and 14% dies” The subject (pregnant women, mothers or expectant mothers) is missing. Also check the subject-verb agreement with “dies”
- Line 76, the phrase “In addition to the midwifery skills” adds very little to message and could be omitted as not all birth attendants are midwives.
- Page 5, lines 90 - 92, the subject appears to late in the sentence. Consider this alternative for example: “This paper aims to assess the competency of healthcare providers at district and sub-district health facilities in northern Bangladesh in managing maternal and newborn complications”

Methodology:

- Page 7, line 165: “The results will be disseminated...” I imagine this has already occurred. I so, please modify appropriately.

	Results:  • Page 7, line 172: consider the heading “Facility delivery service utilization and provision of emergency obstetric and specialized neonatal care:” • Page 9, lines 201 & 203: “A higher proportion of nurses and midwives had ever received any” replace any with “an” • Page 9, lines 202: “in-service training on obstetric complications than” replace than with “compared to” • Table 2, page 9: instead of mean age, authors should consider stating the mean and standard deviation in text and provide here age-group categories • Table 2, Similarly, authors should consider re-categorizing the work experience variable to reflect groups with expected comparable experience and knowledge. For example, < 1 year; 1-2;3-5; >5 • Page 13, line 260, does the “poor” competency refer to the “low” competency level or score? May be better to use the same classification throughout. • Page 14, lines 274 - 275: “were less likely to receive” I recommend that you include the p-value for the distribution. • Page 14, line 292 “it is likely that...” consider replacing likely with “understandable or not surprising”
--	---

REVIEWER	Beena Varghese Public Health Foundation of India, India
REVIEW RETURNED	26-Apr-2019

GENERAL COMMENTS	Very well written article on an important topic. My only suggestion would be that it may be useful for the authors to add in some information about the maternal and newborn outcomes at these facilities and discuss how they match with these findings. Given the low scores among nurses and doctors, it appears that there is a major gap in skills and knowledge. However, having some additional information re outcomes at these facilities would help the reader understand if the lacunae in skills translate to outcomes or not. Agree with the authors conclusion that there is surely need to improve skills and knowledge of providers through skills and emergency drills training. We had done a pilot study showing the value of skills and drills for improving emergency obstetric and newborn care in Karnataka.
---

REVIEWER	Margareth Crisóstomo Portela Oswaldo Cruz Foundation Brazil
REVIEW RETURNED	15-Jun-2019

GENERAL COMMENTS	In the manuscript as a whole, English should be carefully reviewed. The following periods/sentences need to be rewritten/clarified: lines 31-33; 63-64; 67-69 (“dies?”), 73-76 (“Although?”), 201-203, 262-264, 287-289. In line 273, the word “Data” is plural.
---

	I recommend replace “providers” for “healthcare professionals” Healthcare services are also providers, and the choice made in the manuscript may be confusing. I think that references 19 and 20 should be in the background. Independently of focusing on another place/context, what the manuscript adds to the papers published previously? The Kruskal-Wallis test does not exactly compare medians. I suggest the omission of the word “median(s)”, referring only the comparison of scores. In the data analysis subsection of Methodology, it is written that “The detailed methodology of assigning the scores are mentioned elsewhere [19,20] and the authors performed adequate sensitivity analysis to choose between equal or weighted scoring”. This information is insufficient for the reader to know what was done, and the issue is not mentioned again in the manuscript. Additionally, it is important to explore more deeply or critically the findings that previous training made no difference, what was in some extent treated in lines 273-279. How the conclusion in favor of training “especially nurses and midwives” interact with these findings? The area of healthcare quality improvement has emphasized the importance of multi professional work/training. Isn't the conclusion in opposition to this trend?
--	--

VERSION 1 – AUTHOR RESPONSE

Response to the reviewers' comments

Reviewer#1:

This paper applies tailored clinical vignettes to assess the competencies of “skilled birth attendants” at primary and secondary health care facilities in northern Bangladesh to identify and manage some common life-threatening obstetric and newborn complications. Findings highlights the need for local and national health authorities to prioritize continuous professional development and practical clinical trainings to assure a fully competent and skilled birth attendance as we aim to end preventable maternal and newborn deaths in such low-resource settings. The authors list some strengths and limitations of this study; however, authors would need to consider and address some of the following to improve the manuscript and make it publishable:

Major changes

Introduction:

1. Page 4 lines 76 – 79: in defining the role of a skilled birth attendant, authors omit the relevant reference (Making pregnancy safer: the critical role of the skilled attendant: a joint statement by WHO, ICM and FIGO. Geneva: WHO; 2004) that defines who a skilled birth attendant is and their required skills and abilities.

Response: The reference is added in line 86, page 4 (reference #12).

2. Page 3, Lines 80 -89: This section is verbose, and the main argument is difficult to follow. Please make it more concise and clear

Response: We have revised the section and made the argument based on the need for assessing competence of health workers in low income setting using clinical vignettes (line 80-88, page 4).

3. Lines 87-88, authors state some pros for using clinical vignettes but do not mention other valid methods that have been used. For example, in; Harvey SA, Blandón YC, McCaw-Binns A, et al. Are skilled birth attendants really skilled? A measurement method, some disturbing results and a potential way forward. Bull World Health Organ. 2007;85(10):783-90. Reviewing this method may highlight some other limitations of a quick vignette system.

Response: Thank you for your feedback. We have added the following line "Assessment based on clinical vignettes is more feasible and accurate than direct clinical observation of health professional's performance where actual performance could be limited by shortage of essential drugs, equipment or supplies and infrequent presentation of critical obstetric cases (<15% of all pregnancies" referring from this article (reference#20) in line 92-95, page 4. We also compared vignette-based assessment with direct clinical observation or assessment based on chart documentation. We have also added some cons of using vignette in limitation section in line 380-383, page 16.

4. Methodology:

5. Page 6, lines 125 -131; Were all four vignettes read out in full before the practitioner was made to answer? Or it was done one scenario at a time up to where action was required? Please provide a clearer description of how this was done

Response: It was done one scenario at a time up to where action was required. We have revised and elaborated the detail process of recording responses in line 141-147, page 6.

6. Also make clear how the answers were matched as stated in lines 127-129

Response: At first, the research physician read out the scenarios to the provider and asked for specific actions to be taken for each of the scenarios. It was done one scenario at a time up to where action was required. Responses made by the health professionals to each of the scenarios were then matched with the pre-defined vignette action points in the check-list (Appendix S1). The vignette action points were based on the WHO's Integrated Management of Pregnancy and Childbirth (IMPAC) guidelines for the selected obstetric and neonatal cases. Responses were considered as correct if they were in agreement to that in the list, if not exactly matched (line 141-147, page 6).

In line 139-140, page 6, we have corrected "questionnaire" to "questionnaire including a check-list" as the data collection instrument used by the research physician to record the responses made by the health workers.

7. Very little information is provided on how the overall scores were obtained. Although authors refer to two papers with a detailed description, sufficient information must be provided in this paper to enable readers to understand the main indicator of this manuscript without having to read two other papers.

Response: Thank you for the feedback. We have added the following lines elaborating the scoring system for each vignette action points “The authors mentioned in their papers that they had asked expert opinion on the relative importance of the actions for all four vignette cases to assign weighted scores for each action point. Based on the post-hoc analysis the authors decided to keep the simpler approach of assigning equal score for maternal vignettes (case A and B) as there were no major differences in results between equal or weighted scores, however, chose weighted scoring for neonatal vignettes (case C and D)” (line 168-173, page 7). We have also updated the Appendix S1 with the scores for each vignette action points used in this paper.

Results:

8. Page 8 line 174, since readiness and actual provision are not exactly the same, I recommend that you replace the readiness to provision. For example, a facility may be equipped to perform assisted vaginal deliveries and thus is technically ready but may never actually perform same.

Response: We agree with your comment and replaced the word ‘readiness’ to ‘service provision’ in line 212, page 8.

9. The number of facility deliveries is presented as an ordinal variable; however, the basis of the categorization is unclear and not mentioned in the methods section. Furthermore, instead of stating that the average number of deliveries was less than 600 at a particular facility, it may be better to provide the actual average with standard deviation as well as the ranges for such a variable.

Response: Thank you for your feedback. After the revision, average number of annual deliveries conducted in the health facilities is presented in mean with standard deviation (line 218-220, page 8). We have also added the following sentences in methodology for further clarification of the categorization, ‘Average number of annual deliveries conducted in the health facilities is presented in mean with standard deviation. Based on the average case load of deliveries, the health facilities were further categorized into ‘low’ utilization if they conducted less than 600 deliveries a month on an average, ‘medium’ (600-900 deliveries), and ‘high’ (more than 900 deliveries a month). The cut-off points were set roughly matching with the expected number of institutional births in EmONC equipped health facilities based on population size and expected number of pregnancies in Bangladesh (Bangladesh Health Bulletin 2017), however, the range does not represent the national or regional estimate of institutional deliveries in district or sub-district hospitals” (line 183-191, page 8)

10. Similarly, on Page 9, lines 195 -196: Better to provide the mean age with standard deviation and range

Response: We have revised it in line 235-237, page 10.

11. Table 2, Page 9, line 206: It is unclear whether the numbers in the table represent absolute numbers or percentages. Please indicate in the column headings

Response: We have revised it in line 230, table 1, page 9.

12. Page 10, lines 209 -216, It appears also that the vignettes scores were categorized as an ordinal variable. However, no mention of this of the basis is provided in the methods section.

Response: We have added this, "The scores were further categorized into 'high' if the health professional received more than 75% of the total score, 'moderate' for 50-75% of the total, and 'low' for less than 50% of the total." In line 176-178, page 7.

Minor changes

Introduction:

13. Page 4 line 67 "A WHO systematic review shows around 27% dies of haemorrhage and 14% dies" The subject (pregnant women, mothers or expectant mothers) is missing. Also check the subject-verb agreement with "dies"

Response: We have revised it in line 73-75, page 4.

14. Line 76, the phrase "In addition to the midwifery skills" adds very little to message and could be omitted as not all birth attendants are midwives.

Response: We have omitted the phrase while revising the section to respond to comment#2 (from line 82-88, page 4).

15. Page 5, lines 90 - 92, the subject appears too late in the sentence. Consider this alternative for example: "This paper aims to assess the competency of healthcare providers at district and sub-district health facilities in northern Bangladesh in managing maternal and newborn complications"

Response: We have revised according to the suggestion in line 95-97, page 5.

Methodology:

16. Page 7, line 165: "The results will be disseminated..." I imagine this has already occurred. I so, please modify appropriately.

Response: We have revised according to the suggestion in line 202-204, page 8.

Results:

17. Page 7, line 172: consider the heading "Facility delivery service utilization and provision of emergency obstetric and specialized neonatal care:"

Response: We have revised according to the suggestion in line 210-211, page 8.

18. Page 9, lines 201 & 203: "A higher proportion of nurses and midwives had ever received any" replace any with "an"

Response: We have revised according to the suggestion in line 242 and 244, page 10.

19. Page 9, lines 202: "in-service training on obstetric complications than" replace than with "compared to"

Response: We have revised according to the suggestion in line 243, page 10.

20. Table 2, page 9: instead of mean age, authors should consider stating the mean and standard deviation in text and provide here age-group categories

Response: We have revised according to the suggestion, line 247, table 2, page 10.

21. Table 2, Similarly, authors should consider re-categorizing the work experience variable to reflect groups with expected comparable experience and knowledge. For example, < 1 year; 1-2;3-5; >5

Response: We already had the years of work experience as categorical variable, however, we split the 1-5 years into two categories (1-2 and 2-3) as the reviewer suggested, line 247, table 2, page 11.

Discussion:

22. Page 13, line 260, does the "poor" competency refer to the "low" competency level or score? May be better to use the same classification throughout.

Response: We have revised according to the suggestion, line 307, page 14.

23. Page 14, lines 274 - 275: "were less likely to receive" I recommend that you include the p-value for the distribution.

Response: We have revised the line "The data show that the proportion of health professionals receiving a training within the last 12 months on management of maternal complication was lower than those receiving training on neonatal complications (7% vs. 30%) ..." (line 322-325, page 14) and omitted "less likely". It is possible that some of the health professionals received both trainings, therefore, we did not do statistical test to compare the proportions and did not report the p-value.

24. Page 14, line 292 "it is likely that..." consider replacing likely with "understandable or not surprising"

Response: We have revised according to the suggestion in line 348, page 15.

Reviewer#2:

25. Very well written article on an important topic. My only suggestion would be that it may be useful for the authors to add in some information about the maternal and newborn outcomes at these facilities and discuss how they match with these findings. Given the low scores among nurses and doctors, it appears that there is a major gap in skills and knowledge. However, having some additional information regarding outcomes at these facilities would help the reader understand if the lacunae in skills translate to outcomes or not.

Response: We appreciate reviewer's suggestion to highlight the maternal and neonatal health outcomes in these health facilities. Finding association of knowledge and skill scores i.e. competency score according to our paper with health outcome was beyond our research objectives, therefore, we did not collect any data on these health outcomes (e.g. birth outcome). We have added a discussion on this as part of the study limitation and suggested that further research should focus on validating the vignette-based findings by examining the association of health professionals' competency with their actual practices and the related health outcomes for mother and newborn in the health facilities (line 383-388, page 16).

26. Agree with the authors conclusion that there is surely need to improve skills and knowledge of providers through skills and emergency drills training. We had done a pilot study showing the value of skills and drills for improving emergency obstetric and newborn care in Karnataka.

Response: We have studied the paper on the pilot study that the reviewer suggested and find it resourceful for our paper. We have cited the paper (reference#37) where we discussed about the need for skill and drill in discussion section (line 329-334, page 15).

Reviewer#3:

27. In the manuscript as a whole, English should be carefully reviewed.

The following periods/sentences need to be rewritten/clarified: lines 31-33; 63-64; 67-69 ("dies"?), 73-76 ("Although"?), 201-203, 262-264, 287-289.

Response: We have revised according to the suggestion (see line 32-34, page 2; line 69-71, page 4; line 72-75 (dies), page 4; line 82-86 (although), page 4; line 242-244, page 10; line 309-312, page 10; and line 343-345, page 10.

28. In line 273, the word "Data" is plural.

Response: We have revised according to the suggestion in line 322, page 14.

29. I recommend replace "providers" for "healthcare professionals" Healthcare services are also providers, and the choice made in the manuscript may be confusing.

Response: We have revised according to the suggestion throughout the paper and changed it to health care professionals.

30. I think that references 19 and 20 should be in the background. Independently of focusing on another place/context, what the manuscript adds to the papers published previously?

Response: We have revised the background section to discuss about the two papers from which we adopted the clinical vignettes for our paper (line 97-102, page 4-5). The cited papers (reference# 21 & 22), one on maternal health and another on neonatal health, are based on a quality of care evaluation of hospital based MNH services in Ghana. The authors found overall moderate competency of MNH workers and its association with demand for facility deliveries, availability of infrastructure, and workload, however, did not explore the association with different attributes of health care professionals in detail. Independent of focusing on another LMIC context, our paper also explores whether individual-level factors such as cadre type, training on emergency obstetric and neonatal care, work experience and health facility-level factors such as health facility type, obstetric caseloads i.e. annual delivery conducted in health facilities, and provision of emergency obstetric and neonatal care are associated with the competence of the health professionals.

Moreover, while the Ghana study was conducted in all tiers of hospitals among MNH professionals, we focused our study only on those professionals who are designated to provide emergency obstetric services at district and sub-district hospitals in northern region of Bangladesh.

31. The Kruskal-Wallis test does not exactly compare medians. I suggest the omission of the word “median(s)”, referring only the comparison of scores.

Response: We have omitted the word “median” according to the suggestion from line 175, page 7.

32. In the data analysis subsection of Methodology, it is written that “The detailed methodology of assigning the scores are mentioned elsewhere [19,20] and the authors performed adequate sensitivity analysis to choose between equal or weighted scoring”. This information is insufficient for the reader to know what was done, and the issue is not mentioned again in the manuscript.

Response: We have added detail information on our methodology, scoring of vignette cases, and clarified why the equal or weighted scoring was considered by the authors of the papers cited (line 168-173, page 7). We also have addressed a similar comment from reviewer#1 (comment#7).

33. Additionally, it is important to explore more deeply or critically the findings that previous training made no difference, what was in some extent treated in lines 273-279. How the conclusion in favor of training “especially nurses and midwives” interact with these findings? The area of healthcare quality improvement has emphasized the importance of multi professional work/training. Isn't the conclusion in opposition to this trend?

Response: Thank you for your feedback. We have omitted “especially nurses and midwives” from the abstract where it was mentioned and revised our original conclusion accordingly (line 49-51, page 2).

We agree that health care quality improvement initiative requires multi-professional training and we did not recommend providing training to nurses and midwives only, rather suggested to provide training to MNH workers more frequently based on skill-and-drill (line 338-340, page 15). We found that in places where the MNH workers were more exposed to critical cases, they had better competency to manage maternal and neonatal complications, therefore, we propose to conduct those trainings in neighboring high utilization facilities where the original work place does not provide adequate critical cases (line 340-343, page 15).

Regarding the findings that 'previous training made no difference', we already discussed about the need for refresher's training and periodic supervision (line 325-329, page 14-15), however, added the discussion about the need for bringing system-level change to create an enabling environment for the MNH workers to adequately translate their learning from training into actual practice in line 332-334 and 343-345, page 15. We also cited one paper from India (reference# 27) which looked at the effect of skill-and-drill based training on knowledge and skill of MNH workers and explored the facility level and provider level barriers to implement the improved knowledge into practice.